# Discovery of New Uracil and Thiouracil Derivatives as Potential HDAC Inhibitors

**DOI:** 10.3390/ph16070966

**Published:** 2023-07-06

**Authors:** Omnia R. Elbatrawy, Mohamed Hagras, Moshira A. El Deeb, Fatimah Agili, Maghawry Hegazy, Ahmed A. El-Husseiny, Mahmoud Mohamed Mokhtar, Samy Y. Elkhawaga, Ibrahim H. Eissa, Samar El-Kalyoubi

**Affiliations:** 1Pharmaceutical Organic Chemistry Department, Faculty of Pharmacy (Girls), Al-Azhar University, Cairo 11823, Egypt; omniaelbatrawy2@gmail.com (O.R.E.); moshiraadeleldeeb.pharmg@azhar.edu.eg (M.A.E.D.); 2Pharmaceutical Organic Chemistry Department, Faculty of Pharmacy (Boys), Al-Azhar University, Cairo 11884, Egypt; 3Chemistry Department, Faculty of Science (Female Section), Jazan University, Jazan 82621, Saudi Arabia; fatmah2000@gmail.com; 4Biochemistry and Molecular Biology Department, Faculty of Pharmacy (Boys), Al-Azhar University, Cairo 11884, Egypt; drmegawry@azhar.edu.eg (M.H.); ahmedabdullah1984@azhar.edu.eg (A.A.E.-H.); mahmoud.mokhtar@azhar.edu.eg (M.M.M.); syussuf2012@azhar.edu.eg (S.Y.E.); 5Department of Biochemistry, Faculty of Pharmacy, Egyptian Russian University, Badr City 11829, Cairo, Egypt; 6Pharmaceutical Medicinal Chemistry & Drug Design Department, Faculty of Pharmacy (Boys), Al-Azhar University, Cairo 11884, Egypt; 7Department of Pharmaceutical Organic Chemistry, Faculty of Pharmacy, Port Said University, Port Said 42511, Egypt

**Keywords:** anticancer, in silico studies, HDAC inhibitors, thiouracil, uuracil

## Abstract

**Background**: Histone deacetylase inhibitors (HDACIs) are a relatively new class of potential drugs for treating cancer. **Aim**: Discovery of new anticancer agents targeting HDAC. **Methods**: New uracil and thiouracil derivatives panels were designed and synthesized as HDAC inhibitors. The synthesized compounds were tested against MCF-7, HepG2, and HCT-116. HDAC1 and HDAC4 inhibitory activities of these compounds were tested. The most active member was tested for its potential against cell cycle, apoptosis, caspase-3, and caspase-8. Docking studies were carried out against HDAC1. **Results**: Compounds **5a**, **5b**, **5f**, **5i**, **5k**, and **5m** exhibited promising cytotoxic activities. HDAC1 and HDAC4 inhibitory activities of these compounds were tested. Regarding the HDAC1 inhibitory activity, compound **5m** was the most potent member (IC_50_ = 0.05 µg/mL) compared to trichostatin A (IC_50_ = 0.0349 µg/mL). For HDAC4, compound **5m** showed superior activity (IC_50_ = 2.83 µg/mL) than trichostatin A (IC_50_ = 3.349 µg/mL)**.** Compound **5m** showed a high potential to arrest the HCT116 cell cycle at the G0-G1 phase. In addition, it showed an almost 17 times apoptotic effect (37.59%) compared to the control cells (2.17%). Furthermore, Compound **5m** showed significant increases in the levels of caspase-3 and caspase-8. Finally, the uracil and thiouracil derivatives showed accepted binding mods against HDAC. **Conclusions**: Compound **5m** has potential anticancer activity targeting HDAC with a significant apoptotic effect.

## 1. Introduction

Cancer continues to be one of humanity’s most significant public health issues, despite the enormous efforts to combat it [1]. According to the WHO, cancer was the leading cause of death worldwide in 2020, accounting for almost 10 million deaths, or about one in every six [2]. As a result, cancer is viewed as a serious issue for both the economy of nations and the economies of individuals [3,4]. Examples of how the complication of cancer pathology manifests itself include oncogenic mutations, multi-drug resistance, and the activation of compensatory mechanisms [5,6,7]. Finding anticancer options that are more potent and less harmful, based on the various biological and molecular characteristics of cancer pathogenesis, is therefore crucial.

One of the main epigenetic pathways implicated in cancer development is histone acetylation [8]. Histone acetylation is a required precursor to other processes of epigenetic modifications, such as methylation or phosphorylation, and it not only causes genetic alterations on its own [9,10]. Histone acetyltransferases (HAT) and histone deacetylases (HDAC) are two antagonistic categories of enzymes that control the process of histone acetylation. As lysine residues on histone and non-histone proteins are acetylated, heterochromatin is transformed into euchromatin, whereas HDACs play the opposite role by deacetylating chromatin to return it to its more condensed condition [11]

A relatively emerging family of prospective medications for treating hyperproliferative illnesses is histone deacetylase inhibitors (HDACIs) [12,13,14]. These inhibitors bind directly to the HDAC active site and block substrate access, producing an accumulation of acetylated histone [15,16,17]. They can affect differentiation, growth arrest, and/or apoptosis in transformed cell cultures due to their diverse biological activities [18,19]. There is a high demand for novel HDACIs as HDACs have become a key tactic in anticancer drug research [20,21]. Some families of tiny, powerful HDACIs have recently been discovered (Figure 1) [22,23].

HDACIs should have a cap group, a spacer, and a functional group as basic pharmacophores [24]. The reported functional groups are hydroxamic acids, carboxylic acids, and phenylene diamines [25].

Besides, uracil and thiouracil moieties are important N-containing heterocycles in medicinal chemistry and drug discovery [26] due to their wide scope of biological activity [26], especially antitumor activities [27]. So, our goal is the design and synthesis of new uracil and thiouracil-containing derivatives targeting HDAC with promising effects against cancer.

### Rationale of Molecular Design

Studying the SAR of the HDAC inhibitors class revealed three pharmacophoric features essential for maximal fitting in the active site of HDAC. These features include (i) a zinc-binding region group (ZBG) which can interact with the zinc atom in the active site, (ii) a linker moiety that can occupy the tubular access of the active site, and (iii) a cap group which can occupy the surface recognition motif [28] (Figure 2).

In this work, we aimed to synthesize new compounds targeting HDAC. The new compounds were designed to possess the pharmacophoric features of HDAC inhibitors. Many derivatives were applied in this work to reach a good insight into the SAR of the synthesized compounds as potential anticancer agents.

The designed compounds varied in their different pharmacophoric features. For the zinc-binding region, two bioisosters were used. These isosters are the substituted thiouracil (**5a**–**g** and **6a**–**d**) and uracil (**5h**–**m** and **7a**–**c**) derivatives. Different benzyl derivatives (**5a**–**g** and **5i**–**m**) were used as linkers. In one compound (**5h**), a methylene group was used as a linker. Additional series (**6a**–**d** and **7a**–**c**) comprise different cyclic structures as a linker moiety. Regarding the cap group, it was thiouracil (**5a**–**g** and **6a**–**d**) and uracil (**5h**–**m** and **7a**–**c**) derivatives (Figure 3).

## 2. Results and Discussion

### 2.1. Chemistry

The ability of 6-aminouracil to react with aliphatic or aromatic carbonyl compounds [29,30,31,32,33,34,35] has been amply demonstrated up till now. The search for new biologically active substances has fueled interest in these reactions, as the presence of a uracil moiety that is a pharmacophore in an organic molecule frequently provides the molecule with some kind of biological effect. Significant progress has been made in uracil derivatives in the field of chemistry. Our protocol is directed towards synthesized novel derivatives of 5,5′-(arylmethylene)bis(6-aminouracils) and dipyrimidopyridines. 6-Amino-2-oxo(thioxo)pyrimidine-4-ones **3a**,**b** was used as starting material and prepared according to the reported methods [29,30,31,32,33,34] as shown in Figure 1.

The condensation of appropriate aliphatic or aromatic aldehydes (at the ratio carbonyl compound to aminouracils 1:2) with compounds **3a**,**b** in ethanol in the presence of HCl at room temperature resulting in 5,5′-bisdiaminopyrimidines **5a**–**m** in good yields. The latter compounds undergo intermolecular dehydration, as shown in Figure 4. The desired compounds **5a**–**m** were approved by spectral data ^1^H-, ^13^C NMR, IR, and Mass spectra. ^1^H NMR spectra of compounds **5a**–**m** showed a characteristic singlet signals of CH-5 at the range of δ 5.37–6.05 ppm, as well as broad singlet signals characteristic for the 2NH_2_-6 group at δ 7.05–7.95 ppm.

Dipyrimidopyridine derivatives **6a**–**d** were synthesized in good yields from refluxing of compounds **5a**, **f**, **h**, **l** with a mixture of AcOH/c.HCl for 2.5 h (Figure 2). The reaction proceeds through the same idea of the Hantzsch reaction via intramolecular cyclization accompanied by the evolution of NH_3_ due to the attack of the amino group of one unit to the electrophilic carbon center of C=NH, as illustrated in Figure 4. The novel compounds were revealed by ^1^H NMR, ^13^C NMR, IR, and Mass spectra. ^1^H NMR of compounds **6a**–**c** exhibit the disappearance of the four singlet signal protons characteristic for 2NH_2_ groups and the appearance of a singlet signal of NH-10 at δ 7.77–7.37 ppm and another singlet signal characteristic for CH-5 at δ 5.46–5.75 ppm. Moreover, compound **6d** showed a characteristic singlet signal for NH-10 at δ 7.18 and CH_2_-5 at δ 3.49 ppm. A characteristic C-5 signal at the δ 80–90 ppm range was noticed in ^13^C NMR for the mentioned compounds.

On the other hand, Refluxing compounds **5c**, **e**, **g** with a mixture of AcOH/c.HCl for 4–5 h take place through intramolecular oxidative cyclization afforded **7a**–**c** (Figure 2), which is proved by all spectral data. ^1^H NMR proved, without doubt, the formation of oxidative cyclized compounds **7a**–**c** via the disappearance of the singlet signal of NH-10 at the region of δ 7 ppm as well as the clearance of the spectra from the singlet signal of CH-5 at the range δ 5.50–5.70 ppm. Moreover, the characteristic C-5 signal appeared at the normal deshielded aromatic region in ^13^C NMR, which proved the complete aromatization of the pyridine ring. A plausible reaction mechanism might be illustrated as follows in Figure 4.

### 2.2. Biological Testing

#### 2.2.1. In Vitro Cytotoxic Activities

Anti-proliferative effect of the target compounds was assessed against a panel of tumor cell lines, including MCF-7 (human breast cancer cell line), HepG2 (human liver carcinoma cell line), colorectal carcinoma (HCT-116) using MTT assay [36]. Sorafenib was used as a reference drug. From the results presented in Table 1, it is clear that **5a**, **5g**, and **5f** has promising anti-proliferative effect against MCF-7 with IC_50_ values of 11 ± 1.6, 21 ± 2.2, 9.3 ± 3.4 µM, respectively compared to sorafenib (IC_50_ = 141 ± 3 µM). In addition, compound **5b** showed high activity against HCT-116 cells with an IC_50_ value of 21 ± 2.4 µM compared to sorafenib (IC_50_ = 177 ± 0.93 µM). Furthermore, compounds **5i**, **5k**, and **5m** exhibited high cytotoxic effect against HepG2 with IC_50_ values of 4 ± 1, 5 ± 2, and 3.3 ± 0.56 µM, respectively, compared to sorafenib (IC_50_ = 17 ± 2.3 µM).

Some compounds showed moderate activities against MCF-7 as **5c**, **5b**, **5d**, **5e**, **5j** and **5m** with IC50 values of 77 ± 2.3, 55 ± 2.8, 62 ± 2.1, 60 ± 0.49, 71 ± 2, and 52 ± 3.5 µM, respectively. Additionally, compounds **5a**, **5c**, **5e**, **5f**, **5g**, **5h**, and **5i** showed moderate cytotoxic activity against HCT-116 cells with IC_50_ values ranging from 88 ± 2.4 to 97 ± 1.2 µM. On the other hand, the other compounds showed weak activities against the tested cell lines.

#### 2.2.2. Structure-Activity Relationship

The cytotoxicity results show that the synthesized compounds with open chain linkers (**5a**–**m**) are more active than those with cyclic linkers (**6a**–**d**, and **7a**–**c**). These results may be explained by the flexibility of the open chain linkers, which may give a good chance for flexible orientations in the active pocket of the target enzyme. On the other hand, the cyclic linker may restrict the good fitting with the active site of the receptor.

For the cap group, it was found that uracil derivatives are more active than thiouracil derivatives. These findings may be due to the high chance of the oxygen atom of uracil moiety to form electrostatic attraction at the cap-binding region. On the other hand, the sulfur atom of the thiouracil moiety has less chance to form electrostatic bonds than the uracil moiety. More clarification about the binding pattern of the synthesized compounds was clarified in the docking section.

Depending on the cytotoxicity against MCF-7, we can reach more details about SAR. For the thiouracil derivatives with open-chain linkers (**5a**–**g**), it was found that activity decreased upon substitution at the linker region as the order of 4-nitrophenyl > phenyl > thiophene > 4-methylphenyl > 4-methoxyphenyl > 2,4-dichlorophenyl > 4-chlorophenyl.

For the uracil derivatives with open chain linkers (**5h**–**m**), it was found that activity decreased upon substitution at the linker region as the order of 4-chlorophenyl > 4-methylphenyl > H > 2,4-dichlorophenyl > phenyl > 4-methoxyphenyl.

For the thiouracil derivatives with cyclic linkers (**6a**,**b**), it was found that the substitution phenyl moiety (**6a**) is more advantageous than the substitution with 4-nitrophenyl moiety (**6b**).

For the uracil derivatives with cyclic linkers (**6c**,**d**), it was found that the unsubstituted cyclic linker (**6d**) is more advantageous than the substituted one with 4-methoxyphenyl moiety (**6c**).

Compared to the activity of the compound, the thiouracil derivatives **6c** (bearing a propyl moiety at both cap and zinc binding group) and **7b** (bearing an ethyl moiety at both cap and zinc binding group) indicated that the substitution with propyl moiety is better for biological activity.

#### 2.2.3. HDAC1 and HDAC4 Inhibitory Assay

HDAC1 plays a critical role in proliferating and senescent cells in culture and young and old tissues in vivo [37]. HDAC1 levels are also essential for regulating apoptosis [38]. Basic and clinical experimental evidence has established that HDAC4 performs various functions [39]. Accordingly, HDAC1 and HDAC4 were selected for testing in this work.

To assess the mechanism of cytotoxicity of the synthesized compounds, HDAC1 and HDAC4 inhibitory activities of the most cytotoxic compounds (**5a**, **5b**, **5f**, **5i**, **5k**, and **5m**) were tested. Trichostatin A was used as a reference compound. The results were summarized as IC_50_ values in Table 2.

In general, as appeared in Table 2, the synthesized compounds have higher selectivity towards HDAC1 than HDAC4. These results match the reported behavior of uracil derivatives against HDAC1 [40].

Regarding the HDAC1 inhibitory activity, compound **5m** was the most potent member (IC_50_ = 0.05 µg/mL) compared to trichostatin A (IC_50_ = 0.035 µg/mL)**.** In addition, compounds **5i** and **5k** showed strong activities with IC_50_ values of 0.146 and 0.23 µg/mL, respectively. Furthermore, compounds **5a**, **5b**, and **5f** showed moderate HDAC1 inhibitory activities with IC_50_ values of 1.34, 1.01, and 1.9 µg/mL, respectively.

For HDAC4 inhibitory activity, compound **5m** showed superior activity (IC_50_ = 2.83 µg/mL) than trichostatin A (IC_50_ = 3.35 µg/mL)**.** In addition, compounds **5a**, **5b**, **5f**, **5i**, and **5k** showed moderate HDAC4 inhibitory activities with IC_50_ values ranging from 5.27 to 8.311 µg/mL.

#### 2.2.4. Cell Cycle Analysis

The effect of the most promising compound, **5m,** against the cell cycle was tested in HCT116 cells. The tested cells were subjected to compound **5m** with a concentration of 78 µM (IC_50_ value of compound **5m**) after 72 h. As presented in Table 3 and Figure 5, the percent of HCT-116 treated cells increased at the %G0–G1 phase (55.31) compared to its concentration in the control cells (43.82%). On the contrary, the percentage of HCT-116 cells decreased at the S phase from 41.19 to 34.88%. Similarly, it decreased at the G2/M phase from 15.04% to 9.81%. Such findings revealed that compound **5m** arrested the HCT-116 cell growth at G0–G1 phase.

#### 2.2.5. Apoptosis Analysis

Compound **5m** was tested for apoptotic effect in HCT-116 using Annexin-V/propidium iodide (PI) staining assay. The tested cells were subjected to 78 µM from compound **5m** with an incubation time of 72 h. The results revealed that the apoptotic effect of compound **5m** was almost 17 times (37.59%) more than observed in control cells (2.17%). The early apoptosis increased from 0.43% to 22.36%. The late apoptosis increased from 0.18 to 13.14% (Table 4 and Appendix A).

#### 2.2.6. Caspase-3 and Caspase-8 Determination

Due to the potential effect of both caspase-3 and caspase-8 on the apoptosis pathway [41], the effects of the most active candidate **5m** on the level of caspase-3 and caspase-8 were tested on HCT-116 72 h. As shown in Table 5, Compound **5m** showed significant increases in the levels of caspase-3 and caspase-8 (5- and 2.5-fold, respectively) compared to the control cells. Taking Staurosporine as a positive control, compound **5m** showed slightly less activity against the level of caspase-3 and caspase-8, as deduced from Table 5.

#### 2.2.7. Cytotoxicity against Normal Cell Line

The cytotoxicity of the most promising candidate, **5m** against normal cells (WI-38), was assessed using an MTT assay. Staurosporine was used as a reference molecule. The results are summarized in Table 1.

The results revealed that compound **5m** has very low cytotoxicity against WI-38 cells with an IC_50_ value of 65.67 µM compared with Staurosporine (IC_50_ = 51.48 µM). The obtained results indicated that compound **5m** is safer than Staurosporine.

### 2.3. Docking Studies

All the synthesized compounds were docked against the crystal structure of HDAC1 (PDB ID: 1C3R) using MOE2019 software to reach a good insight into their binding pattern. Trichostatin A (The co-crystallized ligand) was utilized as a reference molecule. The binding pattern of some examples is presented below. The binding free energies (∆G) for all the target molecules against HDAC1 are shown in Table 6.

Trichostatin A exhibited a binding score of −19.11 kcal/mol against HDAC1. The hydroxamic acid group occupied the zinc-binding region forming many hydrogen bonds with Gly140 and Tyr297. Also, the hydroxamic acid group was involved in an electrostatic interaction with zinc ions. Three hydrophobic bonds were formed between the linker chain and Leu265, Phe198, and His170. The surface recognition motif was occupied by the *N,N*-dimethylaniline moiety (Figure 6).

Compound **5b** exhibited a binding mode like that of Trichostatin A, with a binding score of −14.13 kcal/mol. One of the two 6-amino-1-ethyl-2-thioxo-2,3-dihydropyrimidin-4(1*H*)-one moieties occupied the zinc-binding region forming one hydrogen bond withHis131. Such moiety formed three electrostatic bonds with Zn ion, Tyr297, and His132, and two hydrophobic bonds with Phe198 and His132. The chlorobenzene moiety occupied the linker region forming two hydrophobic interactions with Leu265 and Tyr196. The other 6-amino-1-ethyl-2-thioxo-2,3-dihydropyrimidin-4(1*H*)-one moiety occupied the surface recognition region forming one hydrophobic interaction with Leu265 (Figure 7).

Compound **5i** showed a binding energy of −17.35 kcal/mol. The first 6-amino-1-propylpyrimidine-2,4(1*H*,3*H*)-dione moiety occupied the zinc binding region forming two hydrogen bonds with His131 and Asp258. In addition, it formed three electrostatic bonds with Zn ion and His132. Also, it formed two hydrophobic bonds with Cys142 and Leu23. The phenyl moiety occupied the linker region. The other 6-amino-1-propylpyrimidine-2,4(1*H*,3*H*)-dione moiety occupied the surface recognition region forming one hydrogen bond and one hydrophobic interaction with Leu265 (Figure 8).

Compound **5k** showed a binding energy of −15.36 kcal/mol. One 6-amino-1-propylpyrimidine-2,4(1*H*,3*H*)-dione moiety was oriented into the zinc binding region, forming one hydrogen bond with Tyr297. Three electrostatic interactions were formed between the first 6-amino-1-propylpyrimidine-2,4(1*H*,3*H*)-dione moiety and Zn ion, His170, and Phe141. Also, it formed three hydrophobic interactions with Phe141, Phe198, and His132. The central tolyl moiety occupied the linker region forming a hydrophobic interaction with Phe200. The other 6-amino-1-propylpyrimidine-2,4(1*H*,3*H*)-dione moiety occupied the surface recognition region forming three hydrophobic interactions with Leu265, Pro22, and Phe141 (Figure 9).

Compound **5m** showed a binding energy of −14.99 kcal/mol. The first 6-amino-1-propylpyrimidine-2,4(1*H*,3*H*)-dione moiety occupied the zinc binding region forming two hydrogen bonds with Tyr297 and His170. In addition, it formed four electrostatic interactions with Zn ion, Leu265, Gly140, and Phe198. Also, it formed four hydrophobic interactions with Phe141, Phe198, Leu265, and His132. The central 4-chlorophenyl moiety occupied the linker region forming a hydrophobic interaction with Leu265. The other 6-amino-1-propylpyrimidine-2,4(1*H*,3*H*)-dione moiety occupied the surface recognition region forming two hydrophobic interactions with Tyr91 and Phe198. In addition, it formed a hydrogen bond with Tyr91 (Figure 10).

## 3. Conclusions

Twenty uracil and thiouracil derivatives were synthesized as potential inhibitors for HDAC. These compounds were tested for their cytotoxic effect against MCF-7, HepG2, and HCT-116 cell lines. Some compounds showed promising anti-proliferative activities. **5a**, **5b**, and **5f**, has promising anti-proliferative effect against MCF-7 with IC_50_ values of 11 ± 1.6, 55 ± 2.8, 9.3 ± 3.4 µM, respectively compared to sorafenib (IC_50_ = 141 ± 3 µM). In addition, compound **5b** showed high activity against HCT-116 cells with an IC_50_ value of 21 ± 2.4 µM compared to sorafenib (IC_50_ = 177 ± 0.93 µM). Furthermore, compounds **5i**, **5k**, and **5m** exhibited high cytotoxic effect against HepG2 with IC_50_ values of 4 ± 1, 5 ± 2, and 3.3 ± 0.56 µM, respectively, compared to sorafenib (IC_50_ = 17 ± 2.3 µM). SAR study revealed that the synthesized compounds with open chain linkers (**5a**–**m**) are more active than that with cyclic linkers (**6a**–**d**, and **7a**–**c**). Compound **5m** showed an IC_50_ of 0.05 µg/mL against HDAC1 compared to trichostatin A (IC_50_ = 0.0349 µg/mL)**.** Furthermore, compound **5m** showed superior activity (IC_50_ = 2.83 µg/mL) than trichostatin A (IC_50_ = 3.349 µg/mL) against HDAC4**.** The most promising compound **5m** arrested the HCT-116 cell growth at the G0-G1 phase and induced apoptosis by 17-fold compared to the control. In addition, such a compound increased the levels of caspase-3 and caspase-8 in HCT-116 cells. Finally, the docking studies indicated that the synthesized compounds have a binding mode almost like the reference molecule (trichostatin A) against the prospective target (HDAC).

## 4. Experimental

### 4.1. Chemistry

#### 4.1.1. General

All advice used in the synthesis and analysis of the new compounds was presented in Appendix A.

#### 4.1.2. 6-Amino-1-alkyl-2-oxo/thioxo-2,3-dihydropyrimidinones (**3a**,**b**)

It was prepared according to the reported method [29,30,31,32,33,34].

#### 4.1.3. 5,5′-(Arylmethylene)bis(6-amino-1-alkyl-2-oxo/thioxo-2,3-dihydropyrimidinones) (**5a**–**m**)

A mixture of compounds 6-aminouracil and/or thiouracil (**3a** and/or **3b**) (2.0 mmol) and different appropriate aromatic aldehydes (1.0 mmol) (**4a**–**h**) in the presence of conc. hydrochloric acid as a catalyst in absolute ethanol (20 mL) was stirred at room temperature for 2–3 h. The formed precipitate was collected by filtration, washed with methanol, and recrystallized from DMF:ethanol (3:1) afforded the desired compounds (**5a**–**m**) in good yields (Figure 1). The structures of the synthesized compounds were confirmed by ^1^H and ^13^C NMR spectroscopy.

##### 5,5′-(Phenylmethylene)bis(6-amino-1-ethyl-2-thioxo-2,3-dihydropyrimidin-4(1*H*)-one) (**5a**)

White Solid, (yield: 77%), m.p. > 300 °C; HPLC (99.65%); IR (KBr) cm^−1^: 3390, 3102 (NH_2_, NH), 3050 (CH arom.), 2972, 2927 (CH aliph.), 1636 (C=O), 1520 (C=C); ^1^H NMR (400 MHz, DMSO-d_6_) δ 12.25 (s, 2H, 2NH, exchangeable with D_2_O), 7.70 (s, 4H, NH_2,_ exchangeable with D_2_O), 7.23 (t, *J* = 7.5 Hz, 2H, arom.), 7.12 (t, *J* = 7.5 Hz, 1H, arom.), 7.07 (d, *J* = 7.6 Hz, 2H, arom.), 5.51 (s, 1H, CH-5), 4.62–4.30 (m, 4H, 2CH_2_), 1.22 (t, *J* = 6.7 Hz, 6H, 2CH_3_); ^13^C NMR (100 MHz, DMSO-d_6_) δ 174.71, 161.44, 154.62, 138.51, 128.33, 126.89, 125.72, 91.29, 43.64, 34.92, 12.37; MS (70 eV) *m*/*z* (%): 430 (M^+^, 5), 258 (16), 102 (31), 44 (100); Anal. Calcd for C_19_H_22_N_6_O_2_S_2_ (430.55): C, 53.00; H, 5.15; N, 19.52; Found: C, 53.19; H, 5.39; N, 19.78.

##### 5,5′-(*p*-Tolylmethylene)bis(6-amino-1-ethyl-2-thioxo-2,3-dihydropyrimidin-4(1*H*)-one) (**5b**)

White solid, (yield: 63%), m.p. = 239–240 °C; HPLC (99.65%); IR (KBr) cm^−1^; 3376, 3158 (NH_2_, NH), 3070 (CH arom.), 2976, 2928 (CH aliph.), 1661 (C=O), 1565 (C=C), 833 (p-substituted phenyl); ^1^H NMR (400 MHz, DMSO-d_6_) δ 12.42 (s, 2H, NH), 8.04 (d, *J* = 8.2 Hz, 2H, arom.), 7.34 (m, 2H, arom.), 7.05 (s, 4H, 2NH_2_), 6.05 (s, 1H, CH-5), 4.58 (q, *J* = 6.9 Hz, 2H, CH_2_), 4.30 (q, *J*=6.8 Hz, 2H, CH_2_), 2.38 (s, 3H, CH_3_), 1.34 (t, *J* = 6.9 Hz, 3H, CH_3_), 1.13 (t, *J* = 6.8 Hz, 3H, CH_3_); ^13^C NMR (100 MHz, DMSO-d_6_) δ 172.90, 161.91, 152.27, 140.52, 129.59, 128.67, 128.25, 127.81, 126.58, 94.32, 42.78, 32.93, 21.01, 12.13; MS (70 eV) *m*/*z* (%): 444 (M^+^, 39), 375 (44), 274 (100), 169 (50), 62 (39); Anal. Calcd for C_20_H_24_N_6_O_2_S_2_ (444.57): C, 54.03; H, 5.44; N, 18.90; Found: C, 54.21; H, 5.67; N, 19.08.

##### 5,5′-((4-Chlorophenyl)methylene)bis(6-amino-1-ethyl-2-thioxo-2,3-dihydropyrimidin- 4(1*H*)-one) (**5c**)

White solid, (yield: 78%), m.p. > 300 °C; IR (KBr) cm^−1^: 3376, 3153 (NH_2_, NH), 3050 (CH arom.), 2976, 2928 (CH aliph.), 1661 (C=O), 1565 (C=C), 839 ((*p*-substituted phenyl); ^1^H NMR (400 MHz, DMSO-d_6_) δ 12.28 (s, 2H, 2NH), 7.69 (s, 4H, 2NH_2_), 7.27 (d, *J* = 8.5 Hz, 2H, arom.), 7.10 (d, *J* = 8.5 Hz, 2H, arom.), 5.48 (s, 1H, CH-5), 4.57-4.36 (m, 4H, 2CH_2_), 1.23 (t, *J* = 6.7 Hz, 6H, 2CH_3_); ^13^C NMR (100 MHz, DMSO-d_6_) δ 174.29, 164.21, 158.52, 137.26, 129.77, 128.48, 127.71, 87.32, 43.19, 34.08, 11.90; MS (70 eV) *m*/*z* (%): 466 (M^+2^, 4), 464 (M^+^, 10), 333 (12), 292 (95), 190 (20), 44 (100); Anal. Calcd for C_19_H_21_ClN_6_O_2_S_2_ (464.99): C, 49.08; H, 4.55; N, 18.07; Found: C, 49.32; H, 4.69; N, 18.23.

##### 5,5′-((2,4-Dichlorophenyl)methylene)bis(6-amino-1-ethyl-2-thioxo-2,3-dihydro pyrimidin-4(1*H*)-one) (**5d**)

White powder (yield: 60%), m.p. = 272–273 °C; IR (KBr) cm^−1^: 3373, 3145 (NH_2_, NH), 3065 (CH arom.), 2975, 2931 (CH aliph.), 1663 (C=O), 1572 (C=C), 730 (trisubstituted phenyl); ^1^H NMR (400 MHz, DMSO-d_6_) δ 12.29 (s, 2H, 2NH), 7.75 (s, 2H, NH_2_), 7.46 (d, *J* = 1.7 Hz, 1H, arom.), 7.33 (dd, *J* = 8.5, 1.7 Hz, 1H, arom.), 7.28 (d, *J* = 8.5 Hz, 1H, arom.), 7.16 (s, 2H, NH_2_), 5.41 (s, 1H, CH-5), 4.45 (m, 4H, 2CH_2_), 1.22 (t, *J* = 6.8 Hz, 6H, 2CH_3_). ^13^C NMR (100 MHz, DMSO-d_6_) δ 174.26, 164.18, 153.87, 136.32, 133.15, 131.04, 130.17, 128.90, 126.71, 90.57, 43.24, 33.55, 11.89; MS (70 eV) *m*/*z* (%): 503 (M^+^4, 23), 501 (M^+^2, 26), 499 (M^+^, 35), 360 (100), 300 (31), 286 (42), 255 (18); Anal. Calcd for C_19_H_20_Cl_2_N_6_O_2_S_2_ (499.43): C, 45.69; H, 4.04; N, 16.83; Found: C, 45.78; H, 4.21; N, 17.05.

##### 5,5′-((4-Methoxyphenyl)methylene)bis(6-amino-1-ethyl-2-thioxo-2,3-dihydropyrimidin -4(1*H*)-one) (**5e**)

White powder (yield: 67%), m.p. = 260–261 °C; ^1^H NMR (400 MHz, DMSO-d_6_) δ 12.22 (s, 2H, 2NH), 7.95 (s, 2H, NH_2_) 7.74 (s, 2H, NH_2_), 6.96 (d, *J* = 8.6 Hz, 2H, arom.), 6.79 (d, *J* = 8.6 Hz, 2H, arom.), 5.45 (s, 1H, CH-5), 4.45 (m, 4H, 2CH_2_), 3.70 (s, 3H, OCH_3_), 1.22 (t, *J* = 6.8 Hz, 6H, 2CH_3_); ^13^C NMR (100 MHz, DMSO-d_6_) δ 174.20, 162.32, 157.06, 153.88, 129.64, 127.45, 113.28, 91.37, 54.94, 43.11, 35.79, 11.90; MS (70 eV) *m*/*z* (%):460 (M^+^, 30), 392 (41), 222 (30), 129 (100), 56 (40); Anal. Calcd for C_20_H_24_N_6_O_3_S_2_ (460.57): C, 52.16; H, 5.25; N, 18.25; Found: C, 52.40; H, 5.37; N, 18.49.

##### 5,5′-((4-Nitrophenyl)methylene)bis(6-amino-1-ethyl-2-thioxo-2,3-dihydropyrimidin-4(1*H*)-one) (**5f**)

White solid, (yield: 79%) m.p. = 274–275 °C; HPLC (99.00%); IR (KBr) cm^−1^: 3384, 3137 (NH_2_, NH), 3060 (CH arom.), 2975, 2931 (CH aliph.), 1662 (C=O), 1555 (C=C), 1505, 1344 (NO_2_), 849 ((*p*-substituted phenyl); ^1^H NMR (400 MHz, DMSO-d_6_) δ 12.34 (s, 2H, 2NH), 8.10 (d, *J* = 8.7 Hz, 2H, arom.), 7.69 (s, 4H, 2NH_2_), 7.38 (d, *J* = 8.7 Hz, 2H, arom.), 5.58 (s, 1H, CH-5), 4.49 (m, 4H, 2CH_2_), 1.23 (t, *J* = 6.8 Hz, 6H, 2CH_3_); ^13^C NMR (100 MHz, DMSO-d_6_) δ 174.42, 162.32, 154.24, 147.27, 145.44, 127.98, 123.02, 90.60, 43.26, 35.79, 11.87; MS (70 eV) *m*/*z* (%): 475 (M^+^, 20), 425 (71), 372 (85), 343 (100), 297 (20) Anal. Calcd for C_19_H_21_N_7_O_4_S_2_ (475.54): C, 47.99; H, 4.45; N, 20.62; Found: C, 48.17; H, 4.63; N, 20.89.

##### 5,5′-(Thiophen-2-ylmethylene)bis(6-amino-1-ethyl-2-thioxo-2,3-dihydropyrimidin-4(1*H*)-one) (**5g**)

White solid, (yield: 56%) m.p. = 271–272 °C; IR (KBr) cm^−1^: 3380, 3142 (NH_2_, NH), 3030 (CH arom.), 2976, 2924 (CH aliph.), 1630 (C=O), 1566 (C=C); ^1^H NMR (400 MHz, DMSO-d_6_) δ 12.29 (s, 2H, 2NH), 7.87 (br.s, 4H, 2NH_2_), 7.27 (d, 1H, arom.), 6.85 (t, 1H, arom.), 6.65 (s, 1H, arom.), 5.64 (s, 1H, CH-5), 4.50 (m, 4H, 2CH_2_).1.21 (t, *J* = 6.9 Hz, 6H, 2CH_3_); ^13^C NMR (100 MHz, DMSO-d_6_) δ 174.35, 162.34, 153.85, 143.73, 126.28, 123.79, 123.60, 91.51, 43.11, 18.56, 11.86; MS (70 eV) *m*/*z* (%): 436 (M^+^, 7), 264 (58), 171 (90), 44 (100); Anal. Calcd for C_17_H_20_N_6_O_2_S_3_ (436.57): C, 46.77; H, 4.62; N, 19.25; Found: C, 46.98; H, 4.76; N, 19.51.

##### 5,5′-Methylenebis(6-amino-1-propylpyrimidine-2,4(1*H*,3*H*)-dione) (**5h**)

White solid, (yield 49%), m.p. = 256–257 °C; IR (KBr) cm^−1^: 3342, 3133 (NH_2_, NH), 2969, 2938 (CH aliph.), 1677 (C=O), 1590 (C=C); ^1^H NMR (400 MHz, DMSO-d_6_) δ 11.40 (s, 1H, NH), 10.53 (s, 1H, NH), 7.52 (br. s, 2H, NH_2_), 7.16 (s, 2H, NH_2_), 3.64 (m, 4H, 2CH_2_), 2.66 (s, 2H, CH_2_), 1.50 (m, 4H, 2 CH_2_), 0.85 (m, 6H, 2CH_3_); ^13^C NMR (100 MHz, DMSO-d_6_) δ 164.02, 161.53, 150.50, 148.84, 84.97, 80.36, 45.59, 45.23, 20.86, 20.72, 17.75, 11.10, 10.71; MS (70 eV) *m*/*z* (%): 350 (M^+^, 18), 386 (61), 220 (100), 144 (37), 61 (80); Anal. Calcd for C_15_H_22_N_6_O_4_ (350.38): C, 51.42; H, 6.33; N, 23.99; Found: C, 51.67; H, 6.45; N, 23.75.

##### 5,5′-(Phenylmethylene)bis(6-amino-1-propylpyrimidine-2,4(1*H*,3*H*)-dione) (**5i**)

Pale yellow solid, yield (66%), m.p. = 294–295 °C; HPLC (99.58%); IR (KBr) cm^−1^: 3388, 3178 (NH_2_, NH), 3045 (CH arom.), 2969, 2930 (CH aliph.), 1670 (C=O), 1598 (C=C); ^1^H NMR (400 MHz, DMSO-d_6_) δ 10.71 (s, 2H, 2NH), 7.73–7.41 (br s, 4H, 2NH_2_), 7.19 (t, *J* = 7.6 Hz, 2H, arom.), 7.07 (m, 3H, arom.), 5.45 (s, 1H, CH-5), 3.77 (m, 4H, 2CH_2_), 1.56 (m, 4H, 2CH_2_), 0.88 (t, *J* = 7.4 Hz, 6H, 2CH_3_); ^13^C NMR (100 MHz, DMSO-d_6_) δ 150.08, 139.67, 134.59, 129.49, 129.16, 127.63, 126.44, 124.84, 71.90, 42.78, 34.13, 20.70, 10.76; MS (70 eV) *m*/*z* (%): 426 (M^+^, 39), 256 (42), 215 (43), 106 (100), 40 (17); Anal. Calcd for C_21_H_26_N_6_O_4_ (426.48): C, 59.14; H, 6.15; N, 19.71; Found: C, 59.36; H, 6.29; N, 19.87.

##### 5,5′-(*p*-Tolylmethylene)bis(6-amino-1-propylpyrimidine-2,4(1*H*,3*H*)-dione) (**5j**)

Pale yellow solid, (yield 73%), m.p. = 286–287 °C; IR (KBr) cm^−1^: 3387, 3180 (NH_2_, NH), 3049 (CH arom.), 2968, 2934 (CH aliph.), 1694 (C=O), 1560 (C=C), 845 (*p*-substituted phenyl); ^1^H NMR (400 MHz, DMSO-d_6_) δ 10.69 (s, 2H, 2NH), 7.67 (br.s, 2H, NH_2_), 7.39 (br.s, 2H, NH_2_), 7.00 (d, *J* = 8.1 Hz, 2H, arom.), 6.92 (d, *J* = 8.1 Hz, 2H, arom.), 5.40 (s, 1H, CH-5), 3.77 (m, 4H, 2CH_2_), 2.23 (s, 3H, CH_3_), 1.55 (m, 4H, 2CH_2_), 0.88 (t, *J* = 7.3 Hz, 6H, 2CH_3_); ^13^C NMR (100 MHz, DMSO-d_6_) δ 160.73, 150.17, 145.35, 136.52, 133.64, 129.80, 129.67, 128.33, 126.43, 80.12, 42.84, 33.84, 20.76, 20.54, 10.81; MS (70 eV) *m*/*z* (%): 440 (M^+^, 27), 327 (60), 210 (62), 97 (100), 61 (86); Anal. Calcd for C_22_H_28_N_6_O_4_ (440.50): C, 59.99; H, 6.41; N, 19.08; Found: C, 60.17; H, 6.53; N, 19.31.

##### 5,5′-((2,4-Dichlorophenyl)methylene)bis(6-amino-1-propylpyrimidine-2,4(1*H*,3*H*)-dione) (**5k**)

White solid, (yield 58%), m.p. 285–286 °C; HPLC (99.53%); IR (KBr) cm^−1^: 3365, 3176 (NH, NH_2_), 3049 (CH arom.), 2969, 2939 (CH aliph.), 1692 (C=O), 1568 (C=C), 709 (trisubstituted phenyl); ^1^H NMR (400 MHz, _6_ δ 10.91 (s, 1H, NH), 10.66 (s, 1H, NH), 7.42 (br.s, 2H, NH_2_), 7.41 (d, *J* = 2.2 Hz, 1H, arom.), 7.31 (dd, *J* = 8.5, 2.2 Hz, 1H, arom.), 7.24 (d, *J* = 8.5 Hz, 1H, arom.), 7.00 (br.s, 2H, NH_2_), 5.37 (s, 1H, CH-5), 3.86–3.68 (m, 4H, 2CH_2_), 1.56 (m, 4H, 2CH_2_), 0.87 (t, 6H, 2CH_3_); ^13^C NMR (100 MHz, DMSO) δ 163.73, 154.59, 149.93, 137.95, 133.20, 130.56, 130.19, 128.75, 126.45, 86.26, 42.80, 33.30, 20.67, 10.71; MS (70 eV) *m*/*z* (%): 499 (M^+^4, 12), 497 (M^+^2, 36), 495 (M^+^, 13), 345 (100), 257 (57), 135 (66), 59 (66); Anal. Calcd for C_21_H_24_ Cl_2_N_6_O_4_ (495.36): C, 50.92; H, 4.88; N, 16.97; Found: C, 51.14; H, 4.95; N, 17.13.

##### 5,5′-((4-Methoxyphenyl)methylene)bis(6-amino-1-propylpyrimidine-2,4(1*H*,3*H*)-dione) (**5l**)

White solid, (yield 64%), m.p. = 289–290 °C; IR (KBr) cm^−1^: 3376, 3150 (NH, NH_2_), 3042 (CH arom.), 2963, 2923 (CH aliph.), 1688 (C=O), 1598 (C=C), 843 (p-substituted phenyl); ^1^H NMR (400 MHz, DMSO-d_6_) δ 10.78 (s, 1H, NH), 10.62 (s, 1H, NH), 7.87 (d, *J* = 8.5 Hz, 1H, arom.), 7.65 (br.s, 2H, NH_2_), 7.37 (br.s, 2H, NH_2_), 7.13 (d, *J* = 8.5 Hz, 1H, arom.), 6.94 (d, *J* = 8.5 Hz, 1H, arom.), 6.76 (d, *J* = 8.5 Hz, 1H, arom.), 5.39 (s, 1H, CH-5), 3.83 (m, 4H, 2CH_2_), 3.69 (s, 3H, O-CH_3_), 1.54 (m, 4H, 2CH_2_), 0.88 (t, *J* = 7.2 Hz, 6H, 2CH_3_); ^13^C NMR (100 MHz, DMSO-d_6_) δ 164.70, 157.26, 150.57, 132.28, 131.76, 130.13, 127.89, 114.99, 113.54, 87.41, 56.17, 55.38, 43.24, 33.86, 21.18, 11.23. MS (70 eV) *m*/*z* (%): 456 (M^+^, 42), 306 (86), 231 (88), 191 (100), 57 (36); Anal. Calcd for C_22_H_28_N_6_O_5_ (456.50): C, 57.88; H, 6.18; N, 18.41; Found: C, 58.09; H, 6.40; N, 18.49.

##### 5,5′-((4-Chlorophenyl)methylene)bis(6-amino-1-propylpyrimidine-2,4(1*H*,3*H*)-dione) (**5m**)

Pale yellow solid, yield (73%), m.p. = 257–258 °C; HPLC (99.71%); IR (KBr) cm^−1^: 3377, 3189 (NH, NH_2_), 3045 (CH arom.), 2966, 2930 (CH aliph.), 1666 (C=O), 1565 (C=C), 843 (*p*-substituted phenyl); ^1^H NMR (400 MHz, DMSO-d_6_) δ 10.81 (s, 2H, 2NH), 7.64 (br.s, 4H, 2NH_2_), 7.24 (d, *J* = 8.6 Hz, 2H, arom.), 7.07 (d, *J* = 8.6 Hz, 2H, arom.), 5.42 (s, 1H, CH-5), 3.76 (m, 4H, 2CH_2_), 1.56 (m, 4H, 2CH_2_), 0.88 (t, *J* = 7.3 Hz, 6H, 2CH_3_); ^13^C NMR (100 MHz, DMSO-d_6_) δ 162.38, 150.10, 138.89, 128.50, 127.55, 86.57, 42.88, 35.83, 20.72, 10.80; MS (70 eV) *m*/*z* (%): 462 (M^+^2, 34), 460 (M^+^, 39), 385 (51), 370 (100), 269 (42), 102 (40); Anal. Calcd for C_21_H_25_ClN_6_O_4_ (460.92): C, 54.72; H, 5.47; N, 18.23; Found: C, 54.95; H, 5.63; N, 18.50.

#### 4.1.4. 1,9-Dialkyl-2,3,5,8,9,10-hexahydropyrido[2,3-*d*:6,5-*d*′]dipyrimidinones (**6a**–**d**)

Compounds **5a**, **5f**, **5h** or **5l** (0.7 mmol) and glacial acetic acid (5 mL) were heated under reflux in the presence of c. HCl (1 mL) for 2.5 h. After cooling, the reaction mixture was filtered off, washed with ethanol crystallized from DMF and dried in the oven.

##### 1,9-Diethyl-5-phenyl-2,8-dithioxo-2,3,5,8,9,10-hexahydropyrido[2,3-*d*:6,5-*d*′]dipyrimidine-4,6(1*H*,7*H*)-dione (**6a**)

White solid, (yield 75%), m.p. = 255–256 °C; ^1^H NMR (400 MHz, DMSO-d_6_) δ 12.78 (s, 2H, NH), 7.70 (s, 1H, NH), 7.26 (t, *J* = 7.5 Hz, 2H, arom.), 7.14 (m, 3H, arom.), 5.46 (s, 1H, CH-5), 4.41–4.31 (m, 4H, 2CH_2_), 1.25 (t, *J* = 7.0 Hz, 3H, CH_3_), 1.19 (t, *J* = 6.9 Hz, 3H, CH_3_); MS (70 eV) *m*/*z* (%): 413 (M^+^, 59), 352 (32), 207 (49), 157 (100), 40 (54); Anal. Calcd for C_19_H_19_N_5_O_2_S_2_ (413.51): C, 55.19; H, 4.63; N, 16.94; Found: C, 55.42; H, 4.78; N, 16.89.

##### 1,9-Diethyl-5-(4-nitrophenyl)-2,8-dithioxo-2,3,5,8,9,10-hexahydropyrido[2,3-d:6,5-d′]dipyrimidine-4,6(1H,7H)-dione (**6b**)

Pale yellow solid, (yield: 86%), m.p. = 251–252 °C; IR (KBr) cm^−1^: 3137 (NH), 3070 (CH arom.), 2977, 2933 (CH aliph.), 1650 (C=O), 1519 (C=C), 1500, 1342 (NO_2_), 853 (*p*-substituted phenyl); ^1^H NMR (400 MHz, DMSO- d_6_) δ 12.83 (s, 2H, 2NH), 8.17–8.06 (m, 2H, arom.), 7.77 (s, 1H, NH), 7.44 (m, 2H, arom.), 5.57 (s, 1H, CH-5), 4.52–4.35 (m, 4H, 2CH_2_), 1.25 (t, *J* = 7.0 Hz, 3H, CH_3_), 1.19 (t, *J* = 7.0 Hz, 3H, CH_3_); ^13^C NMR (100 MHz, DMSO- d_6_) δ 173.95, 162.27, 152.16, 147.22, 145.82, 128.00, 123.20, 92.93, 43.75, 34.65, 11.53; MS (70 eV) *m*/*z* (%): 458 (M^+^, 31), 343 (31), 238 (41), 139 (33), 83 (100); Anal. Calcd for C_19_H_18_N_6_O_4_S_2_ (458.51): C, 49.77; H, 3.96; N, 18.33; Found: C, 49.98; H, 4.17; N, 18.59.

##### 5-(4-Methoxyphenyl)-1,9-dipropyl-5,10-dihydropyrido[2,3-*d*:6,5-*d*′]dipyrimidine-2,4,6,8(1*H*,3*H*,7*H*,9*H*)-tetraone (**6c**)

Yellow solid, (yield: 62%), m.p. = 180–181 °C; IR (KBr) cm^−1^: 3223 (NH), 3091 (CH arom.), 2962, 2937 (CH aliph.), 1647 (C=O), 1543 (C=C), 835 (p-substituted phenyl); ^1^H NMR (400 MHz, DMSO-d_6_) δ 11.48 (s, 1H, NH), 11.37 (s, 1H, NH), 8.35–8.27 (m, 3H, arom., NH) 7.08-7.05 (m, 2H, arom.), 5.75 (s, 1H, CH-5), 3.87 (s, 3H, CH_3_), 3.76-3.71 (m, 4H, 2CH_2_), 1.57-1.53 (m, 4H, 2CH_2_), 0.87 (t, J = 7.5 Hz, 6H, 2CH_3_); ^13^C NMR (101 MHz, DMSO- d_6_) δ 163.48, 163.37, 161.42, 161.08, 155.84, 150.38, 150.32, 137.46, 125.25, 125.11, 115.55, 115.50, 113.92, 83.69, 55.69, 42.30, 41.74, 20.82, 11.19; MS (70 eV) *m*/*z* (%): 439 (M^+^, 27), 311 (66), 245(100), 138 (95), 75 (49); Anal. Calcd for C_22_H_25_N_5_O_5_ (439.47): C, 60.13; 5.73; N, 15.94; Found: C, 59.94; H, 5.91; N, 16.17.

##### 1,9-Dipropyl-5,10-dihydropyrido[2,3-*d*:6,5-*d*′]dipyrimidine-2,4,6,8 (1*H*,3*H*,7*H*,9H)-tetraone (**6d**)

White solid, (yield: 70%), m.p. = 274–275 °C; IR (KBr) cm^−1^: 3233 (NH), 2969, 2938 (CH aliph.), 1677 (C=O), 1589 (C=C); ^1^H NMR (400 MHz, DMSO-d_6_) δ 11.40 (s, 1H, NH), 10.53 (s, 1H, NH), 7.18 (s, 1H, NH), 3.74– 3.60 (m, 4H, 2CH_2_), 3.49 (s, 2H, CH_2_), 1.56–1.44 (m, 4H, 2 CH_2_), 0.86 (t, *J* = 5.0 Hz, 3H, CH_3_), 0.83 (t, *J* = 5.1 Hz, 3H, CH_3_); ^13^C NMR (100 MHz, DMSO-d_6_) δ 161.55, 150.51,150.29, 148.87, 80.37, 45.62, 45.26, 27.70, 20.88, 20.74, 11.11, 10.73; MS (70 eV) *m*/*z* (%): 333 (M^+^, 42), 215 (70), 185 (100), 84 (66); Anal. Calcd for C_15_H_19_N_5_O_4_ (333.35): C, 54.05; H, 5.75; N, 21.01; Found: C, 54.21; H, 5.89; N, 20.96.

#### 4.1.5. 5Aryl-1,9-Diethyl-2,8-dithioxo-2,3,8,9-tetrahydropyrido[2,3-*d*:6,5-*d*′]dipyrimidine-4,6(1*H*,7*H*)-diones (**7a**–**c**)

A mixture of compound **5c**, **5e** or **5g** (0.7 mmol), glacial acetic acid (5 mL) and c. HCl (1 mL) was heated under reflux for 4–5 h. After cooling, the formed precipitate was collected by filtration, washed with ethanol crystallized from DMF and dried in the oven.

##### 5-(4-Chlorophenyl)-1,9-diethyl-2,8-dithioxo-2,3,8,9-tetrahydropyrido[2,3-*d*:6,5-*d*′]dipyrimidine-4,6(1*H*,7*H*)-dione (**7a**)

Yellow solid, (yield: 72%), m.p. = 224–225 °C; IR (KBr) cm^−1^: 3229 (NH), 3109 (CH arom.), 2976, 2928 (CH aliph.), 1662 (C=O), 1572 (C=C), 834 (*p*-substituted phenyl); ^1^H NMR (400 MHz, DMSO-d_6_) δ 10.01 (s, 2H, 2NH), 7.93 (d, *J* = 8.5 Hz, 2H, arom.), 7.68 (d, *J* = 8.5 Hz, 2H, arom.), 4.32–4.22 (m, 2H, CH_2_), 4.12–4.05 (m, 2H, CH_2_), 1.30–1.10 (m, 6H, 2CH_3_); MS (70 eV) *m*/*z* (%): 448 (M^+^2, 6), 446 (M^+^, 17), 402 (77), 325 (100), 249 (54), 69 (22); Anal. Calcd for C_19_H_16_ClN_5_O_2_S_2_ (445.94): C, 51.17; H, 3.62; N, 15.71; Found: C, 51.37; H, 3.76; N, 15.92.

##### 1,9-Diethyl-5-(4-methoxyphenyl)-2,8-dithioxo-2,3,8,9-tetrahydropyrido[2,3-*d*:6,5-*d*′]dipyrimidine-4,6(1*H*,7*H*)-dione (**7b**)

Mustard yellow, yield (66%), m.p. = 221–221 °C; IR (KBr) cm^−1^: 3235 (NH), 3070 (CH arom.), 2988, 2935 (CH aliph.), 1660 (C=O), 1569 (C=C), 843 (*p*-substituted phenyl); ^1^H NMR (400 MHz, DMSO-d_6_) δ 12.47 (s, 1H, NH), 12.40 (s, 1H, NH), 8.43 (d, *J* = 9.0 Hz, 1H, arom.), 8.31(d, *J* = 9.0 Hz, 1H, arom.), 7.10–7.07 (m, 2H, arom.), 4.32 (q, *J* = 6.9 Hz, 4H, 2 CH_2_), 3.89 (s, 3H, CH_3_), 1.18 (t, *J* = 6.9 Hz, 6H, 2CH_3_); ^13^C NMR (100 MHz, DMSO-d_6_) δ 178.64, 178.57, 164.23, 164.10, 162.52, 157.35, 156.82, 156.67, 131.82, 129.66, 127.46, 114.53, 114.14, 113.12, 55.82, 55.70, 54.89, 12.34, 12.26; MS (70 eV) *m*/*z* (%): 441 (M^+^, 18), 355 (41), 235 (100), 113 (28), 50 (39); Anal. Calcd for C_20_H_19_N_5_O_3_S_2_ (441.52): C, 54.41; H, 4.34; N, 15.86; Found: C, 54.21; H, 4.51; N, 16.04.

##### 1,9-Diethyl-5-(thiophen-2-yl)-2,8-dithioxo-2,3,8,9-tetrahydropyrido[2,3-*d*:6,5-*d*′]dipyrimidine-4,6(1*H*,7*H*)-dione (**7c**)

Brown solid, (yield: 54%), m.p. = 216–217 °C; IR (KBr) cm^−1^: 3207 (NH), 3050 (CH arom.), 2972, 2926 (CH aliph.), 1657 (C=O), 1568 (C=C); ^1^H NMR (400 MHz, DMSO-d_6_) δ 12.50 (s, 2H, 2NH), 8.38 (t, 1H, arom.), 8.27 (t, 1H, arom.), 7.41 (m, 1H, arom.), 4.34 (m, 4H, 2CH_2_), 1.17 (m, 6H, 2CH_3_); ^13^C NMR (100 MHz, DMSO-d_6_) δ 178.59, 178.55, 161.36, 160.34, 160.21, 159.38, 143.60, 136.87, 128.88, 128.79, 111.98, 100.93, 54.86, 48.17, 14.51, 12.27; MS (70 eV) *m*/*z* (%): 417 (M^+^, 37), 399 (100), 298 (25), 122 (58), 44 (40); Anal. Calcd for C_17_H_15_N_5_O_2_S_3_ (417.52): C, 48.90; H, 3.62; N, 16.77; Found: C, 49.08; H, 3.80; N, 16.98.

### 4.2. Biological Testing

#### 4.2.1. In Vitro Cytotoxic Activity

Anti-proliferative activity activities were tested using MTT assay [36,42,43,44,45] as described in Appendix A.

#### 4.2.2. In vitro HDAC Assay

Compounds **5a**, **5b**, **5f**, **5i**, **5k**, and **5m** were tested for their HDAC inhibitory activities (HDAC1 and HDAC4 subtypes) as described in Appendix A [46,47].

#### 4.2.3. Flow Cytometry Analysis for Cell Cycle

Cell cycle analysis was performed for compound **5m** as described in Appendix A [48,49,50].

#### 4.2.4. Flow Cytometry Analysis for Apoptosis

The apoptotic effect of the compound **5m** was tested as described in Appendix A [51,52,53]. Quantitative Real-Time Reverse-Transcriptase PCR (qRT-PCR) technique Using qRT-PCR, the effect of compound 5m on the expression of cleaved caspase-3 and caspase-8 was determined (Appendix A) [54,55,56,57,58].

### 4.3. Docking Studies

Docking studies were conducted against HDAC1 (http://www.rcsb.org/ accessed on 1 January 2023, PDB code: 1C3R, resolution of 2.00 Å) as described in Appendix A [59].

## Data Availability

Not applicable.

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
