# Peer review of "Discovery of New Uracil and Thiouracil Derivatives as Potential HDAC Inhibitors"

_pharmaceuticals, 2023, doi:10.3390/ph16070966_

Round 1

Reviewer 1 Report

The manuscript shows a lot of promise, but there are many major issues that need to be addressed before it can be published. Throughout the manuscript, there are several language mistakes. The paper needs rewriting and thorough language editing to allow for a proper peer review., I recommend a professional round of language editing before the paper is published

The paper needs rewriting and I recommend a professional round of language editing before the paper is published

Author Response

Reviewer 1

The paper needs rewriting and I recommend a professional round of language editing before the paper is published.

Thank you for your efforts and your valuable comments. All comments were considered in high interest and all changes were highlighted in the revised manuscript.

  • Abstract needs to be modified and improved.

Response: Done

  • Introduction needed more with sufficient background information. What is SAR?

Response: additional background was added. The essential pharmacophoric fearures HDACIs were clarified in rational section.

  • References missing at some places

Response. Some additional references were added in the revised manuscript.

  • Results and discussions- It seems to me more of results description. Discussion part missing a lot. Rewriting required.

Response: additional clarifications of the results were included in the revised manuscript.

  • 2.1 section-

-5g compound have IC50 21+ 2.2 μM showing moderate activities and 5b

compound with IC50 55+ 2.8 μM showed promising? Please verify these.

- There are lots of errors in sentence writing.

Response: thank you for this notice. All requested corrected were carried out.

  • 2.4. -The tested cells were subjected to 78 μM μM from compound (of) 5m

with incubation for 72h.

Response: corrected.

  • 2.5.- compound 5m showed slightly less activity against the (as deduced

from) levels of caspase 3 and caspase 8.

-Tables need to be formatted with proper sample ID and treatment. (Instead of

writing 5m/HCT116 and Cont. HCT116, I suggest to write as treated and

untreated)

-μM μM has been repeated at multiple places in the manuscript including the

Supplementary.

Response All these issues were solved in the revised version.

10- 2.3 check exhibited spelling

-Also “with” missing in many paragraphs. Please thoroughly check this.

-Example one hydrophobic interaction…Leu265 (Fig. 7).

-Hydrogen bond…His131 and Asp258…

Response: thank you for this notice. These typing mistakes were corrected.

9- Supplementary sections:

-Need to be rewritten especially methods part as there are many errors and inappropriate sentences like human cancer lines were dropped; the cells were exposed to centrifugation…

-Repetition of μM at many places

Response: Supplementary sections were revised.

Reviewer 2 Report

Comments:

The submitted manuscript describes the discovery of new uracil and thiouracil derivatives as potential HDAC inhibitors. The authors designed and synthesized a series of novel uracil and thiouracil derivatives tested against MCF-7, HepG2, and HCT16, moreover, these synthesized compounds exhibited moderate to highly cytotoxic activities. The docking studies also, revealed that the synthesized compounds have a binding mode almost like that of the reference molecule. In my opinion, the manuscript is presented very clearly and scientifically strong, therefor the manuscript could be publishable in pharmaceuticals.

Author Response

Thank you for your praise and valuable revision

Reviewer 3 Report

The authors Omnia R. Elbatrawy and co-workers submitted the manuscript entitled “Discovery of new uracil and thiouracil derivatives as potential HDAC inhibitors” to the journal “Pharmaceuticals” in order to be considered for publication as an “Article”.

The manuscript describes the chemical synthesis and analytical characterization (1H NMR, 13C NMR, IR, MS, CHN-analysis) of a series of potential HDAC inhibitors. Structurally, the newly synthesized compound differ regarding uracil or thiouracil derivatives and linear or cyclic linkers, respectively. The compounds were investigated regarding their cytotoxicity using three different cancer cell lines (MCF-7, HepG2, HCT-116). The most active representatives were assessed regarding inhibition of HDAC1/4. The most promising compound 5m was subject to cell cycle analysis and apoptosis experiments (flow cytometry, Cas3/8 assay). In addition, docking studies using HDAC were performed with the most active representatives.

Scheme 2 vs. Figure 3: Compounds 7a-c are different. Please revise!

@ Figure 2: Please mention in the caption of the Figure that trichostatin is shown in the schematic figure.

The determination of the cytotoxicity was performed with different cancer cell lines. This is a worthy approach. However, in order to exclude general toxicity, performance with a non-cancer cell line is state of the art and therefore highly recommended.

@ Table 1: Please revise the presentation of the values, e.g., considering significant digits

HDAC1 and HDAC4 inhibitory assay: What was the reason to select these particular isoenzymes, as there are also other HDAC enzymes? The authors are kindly asked to provide explanation(s) in their manuscript.

@ Table 2: It seems that there is a selectivity towards inhibition of isoenzyme HDAC1 compared to HDAC4. The authors should discuss this aspect in their manuscript and provide, if possible, an explanation for this behavior.

“were docked against the crystal structure of HDAC” Please clarify which HDAC.

The authors found some interesting structure-activity-relationships. Maybe it is possible to summarize these findings, if possible using a table, to allow an overview and to guide other researcher to facilitate future design of similar compounds.

4.2.1 @ “using MTT assay protocol [31-34]”, also 4.2.3, 4.2.4, 4.2.5 Is it really necessary to provide more than reference citing the MTT assay/experimental procedure?

Compounds that undergo biological testing should have a purity >95%. However, the authors do not report about the purity of the compounds at all, apart from indirectly via CHN analysis. Can the authors add purity values to their manuscript?

Figure 5 is pixelated and would therefore benefit from a higher resolution.

How about the stability of compounds 5a-g and 5h-m? Compared to compounds 6a-d and 7a-c, formal elimination of ammonia would result in these compounds. The authors should confirm stability, especially under the conditions for the cytotoxicity testing.

There are some formal aspect that would benefit from improvement, such as: punctuation, introduction of abbreviations and consistent use throughout the manuscript, caption, subscript, bold, space with units, with/without dash, presentation of the references (see author guidelines), author contribution/conflict of interest/… (see author guidelines).

All the best!

Moderate editing of English is suggested.

Author Response

The authors Omnia R. Elbatrawy and co-workers submitted the manuscript entitled “Discovery of new uracil and thiouracil derivatives as potential HDAC inhibitors” to the journal “Pharmaceuticals” in order to be considered for publication as an “Article”.

The manuscript describes the chemical synthesis and analytical characterization (1H NMR, 13C NMR, IR, MS, CHN-analysis) of a series of potential HDAC inhibitors. Structurally, the newly synthesized compound differ regarding uracil or thiouracil derivatives and linear or cyclic linkers, respectively. The compounds were investigated regarding their cytotoxicity using three different cancer cell lines (MCF-7, HepG2, HCT-116). The most active representatives were assessed regarding inhibition of HDAC1/4. The most promising compound 5m was subject to cell cycle analysis and apoptosis experiments (flow cytometry, Cas3/8 assay). In addition, docking studies using HDAC were performed with the most active representatives.

Thank you for your praise and valuable comments. I considered these comments with high interest. The comments and my response are summarized in the following points.

  • Scheme 2 vs. Figure 3: Compounds 7a-c are different. Please revise!

Response: Adjusted

  • @ Figure 2: Please mention in the caption of the Figure that trichostatin is shown in the schematic figure.

Response: Done

  • The determination of the cytotoxicity was performed with different cancer cell lines. This is a worthy approach. However, in order to exclude general toxicity, performance with a non-cancer cell line is state of the art and therefore highly recommended.

Response: Cytotoxicity against normal cell line was carried out for the most active member.

  • @ Table 1: Please revise the presentation of the values, e.g., considering significant digits

Response: Done

  • HDAC1 and HDAC4 inhibitory assay: What was the reason to select these particular isoenzymes, as there are also other HDAC enzymes? The authors are kindly asked to provide explanation(s) in their manuscript.

Response: done

  • @ Table 2: It seems that there is a selectivity towards inhibition of isoenzyme HDAC1 compared to HDAC4. The authors should discuss this aspect in their manuscript and provide, if possible, an explanation for this behavior.

Response: Done

  • “were docked against the crystal structure of HDAC” Please clarify which HDAC.

Response: Done

  • The authors found some interesting structure-activity-relationships. Maybe it is possible to summarize these findings, if possible using a table, to allow an overview and to guide other researcher to facilitate future design of similar compounds.

Response: Additional clarifications were added to the SAR studies to be more comprehensive.

  • 2.1 @ “using MTT assay protocol [31-34]”, also 4.2.3, 4.2.4, 4.2.5 Is it really necessary to provide more than reference citing the MTT assay/experimental procedure?

Response: Done

  • Compounds that undergo biological testing should have a purity >95%. However, the authors do not report about the purity of the compounds at all, apart from indirectly via CHN analysis. Can the authors add purity values to their manuscript?

Response: thank you for this recommendation. The purity were further tested using HPLC analysis. See the HPLC chats in Supplementary data.

  • Figure 5 is pixelated and would therefore benefit from a higher resolution.

Response: As requested, the resolution of fig. 5 was improved to be 600 dpi.

  • How about the stability of compounds 5a-g and 5h-m? Compared to compounds 6a-d and 7a-c, formal elimination of ammonia would result in these compounds. The authors should confirm stability, especially under the conditions for the cytotoxicity testing.

Response: Thank you, dear respected Professor for this observation which obviously indicates a careful revision. Elimination of ammonia from compounds 5a-g and 5h-m forming compounds 6a-d and 7a-c is occurred under acidic conditions which has an important role as shown in figure 4 as well as under reflux condition for 2.5 hours to 5 hours, but the cytotoxicity testing was conducted by dissolving the tested compounds in DMSO and all the procedure did not exceeds 37 ËšC. Moreover, the 1H NMR spectra proved the structure with the appearance of the broad four protons of singlet signals characteristic for 2NH2-6 group at δ 7.05-7.95ppm with taking into consideration that the compounds have been dissolved in DMSO as the manner in case of cytotoxicity testing.

  • There are some formal aspect that would benefit from improvement, such as: punctuation, introduction of abbreviations and consistent use throughout the manuscript, caption, subscript, bold, space with units, with/without dash, presentation of the references (see author guidelines), author contribution/conflict of interest/… (see author guidelines).

Response: All these issues were solved in the revised manuscript.

Round 2

Reviewer 1 Report

I would like to thank the authors for addressing my initial comments. I am satisfied with the author’s responses to my questions/issues raised in my initial review. The revised manuscript is easier to follow. However, I recommend that revised paper be accepted with some minor revisions. 

Author Response

Reviewer 1

 I would like to thank the authors for addressing my initial comments. I am satisfied with the author’s responses to my questions/issues raised in my initial review. The revised manuscript is easier to follow. However, I recommend that revised paper be accepted with some minor revisions. My comments/corrections below are:

Main text section

➢ Abstract: caspase3 and caspase8….(caspase 3 and caspase 8 or caspase-3 and caspase-8)

Response: Corrected as caspase-3 and caspase-8 in all manuscript.

➢ 1.1: Studying the SAR….(SAR full form still missing)

Response: More details about SAR study were added in the revised manuscript.

➢ Table 3 and 4…(sample ID and treatment, write as treated and untreated)

Response: Done

➢ 2.2.3: On contrast….(on the contrary)

Response: Done

➢ 2.2.4: Compound 5m, the most promising member,…(remove the most promising member)

Response: Done

Supplementary section

➢ 125 MHz in dimethylsulfoxide (DMSO-d6) and TMS as an internal standard…..(DMSO-D6? TMS?)

Response: corrected

➢ The in vitro antiproliferative activities of all the synthesized compounds against three human tumor cell lines namely, MCF-7 (human breast cancer cell line), HepG2 (human liver carcinoma cell line), colorectal carcinoma (HCT-116) was (“were’’) evaluated quantitatively as described in the literature, using MTT assay protocol. Two commercially available drugs (erlotinib) were used in this test as positive controls….(what are those two drugs)

Response: corrected

➢ Human cancer cell lines were dropped in 96-well plates…. (were seeded)

Response:  Done

➢ Next, the wells were incubated….(cells)

Response: Done

➢ Then, for each well, the growth medium was exchanged with 0.1 ml of fresh medium containing graded concentrations of the test compounds to be or equal DMSO….(please check the sentence)

Response: Done

➢ After that, the cells were collected and washed with PBS two successive times with centrifugation…(“times”)

Response:  checked

➢ analyzed. flowjo software….(Flowjo)

Response: Done

➢ was assessed by qRT-PCR (reference)…(reference missing)

Response: Done

➢ determination of caspase3 and caspase8….(caspase 3 and caspase 8 or caspase-3 and caspase-8)

Response: corrected

➢ [as a normalization (housekeeping) gene]……(as a housekeeping gene)

Response: corrected

➢ cDNA (2 μl aliquots) was mixed with 1 μl of forward primer, 1 μl reverse primer, 10 μl master mixture, and the reaction volume was completed to 20 μl with nuclease-free water. All experiments were performed in triplicates…. (rewriting required)

Response: Done

➢ crystal structure of HDAC1. trichostatin A…..(Trichostatin)

Response: Done

Reviewer 3 Report

The authors submitted a revised version of their manuscript for reconsideration for publication in the journal "Pharmaceuticals".

The authors have made improvements in the manuscript such as chemical structures, additions to caption of figures, insertion of higher resolution figures, testing of cytotoxicity in a nontumorigenic cell line, removal of redundant citations, and proof of purity by HPLC analyses (anyways, see below). Otherwise, the authors have also provided comprehensible explanations.

In the HPLC chromatograms in the Supplementary Material, however, such high values for purity do not seem comprehensible. The authors are asked to provide integrated chromatograms. On page 41 I cannot imagine that the peak of compound 5k (8.0 min) is 99.53% and the impurity from 8.5 min to 10.5 min is only 0.23%. Is the ratio of the areas correct? I cannot imagine it. - Another remark: The retention times have the unit "min".

However, the presentation of the numerical values does not yet correspond to the scientific manner. Please revise this. In addition, there are still some issues of formatting to be revised (e.g. different fonts, puctuation, among others), but this can/will be done in the context of editing or proofreading.

All the best!

Minor editing of English is suggested.

Author Response

Reviewer 2

The authors submitted a revised version of their manuscript for reconsideration for publication in the journal "Pharmaceuticals".

The authors have made improvements in the manuscript such as chemical structures, additions to caption of figures, insertion of higher resolution figures, testing of cytotoxicity in a nontumorigenic cell line, removal of redundant citations, and proof of purity by HPLC analyses (anyways, see below). Otherwise, the authors have also provided comprehensible explanations.

In the HPLC chromatograms in the Supplementary Material, however, such high values for purity do not seem comprehensible. The authors are asked to provide integrated chromatograms. On page 41 I cannot imagine that the peak of compound 5k (8.0 min) is 99.53% and the impurity from 8.5 min to 10.5 min is only 0.23%. Is the ratio of the areas correct? I cannot imagine it. - Another remark: The retention times have the unit "min".

However, the presentation of the numerical values does not yet correspond to the scientific manner. Please revise this. In addition, there are still some issues of formatting to be revised (e.g. different fonts, puctuation, among others), but this can/will be done in the context of editing or proofreading.

Response: Thank you for your valuable revision that increased the consistency of our work.

For the purity of the synthesized compounds, we carried out further crystallization to ensure the high purity. The purity was checked first time using TLC. Then, the spectral data confirmed such purity. In addition, elemental analysis gave additional confirmation.

For HPLC analysis, it was carried out out door because it is not available in our institution. All methods were described in supplementary data.